# Functional Network Connectivity Reveals the Brain Functional Alterations in Breast Cancer Survivors

**DOI:** 10.3390/jcm11030617

**Published:** 2022-01-26

**Authors:** Tatyana Bukkieva, Maria Pospelova, Aleksandr Efimtsev, Olga Fionik, Tatyana Alekseeva, Konstantin Samochernych, Elena Gorbunova, Varvara Krasnikova, Albina Makhanova, Anatoliy Levchuk, Gennadiy Trufanov, Stephanie Combs, Maxim Shevtsov

**Affiliations:** 1Personalized Medicine Centre, Almazov National Medical Research Centre, 2 Akkuratova Str., 197341 Saint Petersburg, Russia; tanya-book25@mail.ru (T.B.); pospelovaml@mail.ru (M.P.); atralf@mail.ru (A.E.); fvolga@mail.ru (O.F.); atmspb@mail.ru (T.A.); neurobaby12@gmail.com (K.S.); lenagorbunova-124@yandex.ru (E.G.); varya.krasnikova.93@mail.ru (V.K.); a.mahanova.a@mail.ru (A.M.); feuerlag999@yandex.ru (A.L.); trufanovge@mail.ru (G.T.); 2Department of Radiation Oncology, Technishe Universität München (TUM), Klinikum rechts der Isar, Ismaningerstr. 22, 81675 Munich, Germany; stephanie.combs@tum.de; 3Laboratory of Biomedical Nanotechnologies, Institute of Cytology of the Russian Academy of Sciences (RAS), Tikhoretsky Ave. 4, 194064 Saint Petersburg, Russia

**Keywords:** breast cancer, breast cancer treatment, post-mastectomy pain syndrome, functional MRI, connectome, default mode network

## Abstract

Different neurological and psychiatric disorders such as vertebrobasilar insufficiency, chronic pain syndrome, anxiety, and depression are observed in more than 90% of patients after treatment for breast cancer and may cause alterations in the functional connectivity of the default mode network. The purpose of the present study is to assess changes in the functional connectivity of the default mode network in patients after breast cancer treatment using resting state functional magnetic resonance imaging (rs-fMRI). Rs-fMRI was performed using a 3.0T MR-scanner on patients (N = 46, women) with neurological disorders (chronic pain, dizziness, headaches, and/or tinnitus) in the late postoperative period (>12 months) after Patey radical mastectomy for breast cancer. According to the intergroup statistical analysis, there were differences in the functional connectivity of the default mode network in all 46 patients after breast cancer treatment compared to the control group (*p* < 0.01). The use of rs-fMRI in in breast cancer survivors allowed us to identify changes in the functional connectivity in the brain caused by neurological disorders, which correlated with a decreased quality of life in these patients. The results indicate the necessity to improve treatment and rehabilitation methods in this group of patients.

## 1. Introduction

Currently, breast cancer is the most common cancer among women worldwide—according to GLOBOCAN, 2.26 million new cases of breast cancer were detected in 2020, which accounted for 11.7% of all cancers [1]. In Russia, breast cancer also occupies the first place in the structure of cancer incidence in women, accounting for about 20.9% of cases [2]. In most cases, breast cancer treatment is complex and usually includes surgical treatment, chemotherapy, and/or radiation therapy. As a rule, it is carried out in three stages: preoperative (induction) chemotherapy, local (surgery ± radiation therapy), and postoperative treatment (adjuvant therapy). One of the most common methods for the surgical treatment of breast cancer is Patey radical mastectomy, which consists of removing the breast, the surrounding fat and lymph nodes, the small pectoral muscle.

After treatment for breast cancer, a complex of symptoms occurs, including disorders of the lymphatic, cardiovascular, musculoskeletal, and nervous systems [3,4]. Currently, special attention is being paid to neuropsychiatric disorders after breast cancer treatment, which present in the form of changes in both the peripheral and central nervous systems. Peripheral neurological disorders after breast cancer treatment are associated with persistent pain syndrome and impaired sensitivity and muscle strength of the upper limb on the side of surgical treatment [5,6]. These changes are primarily caused by disorders of the peripheral nervous system due to local fibrous-atrophic postoperative and post-radiation changes [7,8]. However, at later stages, the “centralization” of chronic pain syndrome occurs with the involvement of structural and functional elements of the “pain connectome” of the brain [9]. In the late postoperative period, patients also suffer from so-called thoracic outlet syndrome [10], which is caused by the compression of the vertebral artery on the side of surgical treatment with hypertrophic scalenus muscles, which leads to vertebral-basilar insufficiency [11,12]. According to recent data, mental disorders including the development of severe depression occur in about 25% of women after breast cancer treatment [13,14]. All of the above changes can lead to structural and functional changes in the brain, which worsens the long-term prognosis of rehabilitation and the quality of life of patients [15].

A modern promising method for assessing functional changes in the brain in breast cancer survivors is functional MRI (fMRI) and is based on the BOLD effect (blood oxygenation level dependent), which allows the activation of various areas of the brain to be determined on the basis of the hemodynamic changes that occur in response to the presentation of a particular stimulus or at rest [16]. The most common variant of fMRI is resting state fMRI (rs-fMRI), which evaluates the functional connectivity between the areas of the brain that make up the so-called resting state networks [17,18]. One of the most important and well-studied resting state networks is the default mode network, which includes extensive areas of the medial prefrontal cortex (MPFC), posterior cingulate cortex, and precuneus [19,20]. DMN is involved in the cognitive processes of memory, attention, and emotion regulation; the role of its functional disorders in the pathogenesis of many neurological and mental diseases, including in chronic pain syndrome, has been proven [21,22,23,24,25].

In the current study, rs-fMRI was employed to allow us to evaluate the changes in the brain connectome—the totality of all of the functional networks of the brain that play a key role in the organization of the central nervous system.

## 2. Materials and Methods

### 2.1. Participants

An open single-center controlled study of the functional connectivity of the default mode network in patients following breast cancer treatment was conducted.

The study enrolled 46 patients (women) aged 35 to 50 years old after breast cancer treatment: surgical treatment (unilateral or bilateral Patey radical mastectomy), a combination of surgical treatment and radiation therapy (local therapy), a combination of surgical treatment and systemic therapy, or complex treatment (combination of surgery, radiation therapy, and chemotherapy), who developed various post-treatment symptoms associated with treatment but not with the primary cancer disease. All of the patients were in the late postoperative period (>12 months) after Patey radical mastectomy. The control group enrolled 20 healthy female volunteers from the same age category.

### 2.2. Exclusion Criteria

The exclusion criteria included signs of main oncological disease progression; the presence of distant breast cancer metastases, including brain metastases, brain tumors, demyelination diseases, brain development anomalies, traumatic brain injury, and other relevant brain pathologies; the presence of hemodynamically significant atherosclerotic stenoses of the main arteries in the head and neck; acute infectious and mental diseases; pregnancy; decompensated somatic pathology; and contraindications to MRI.

All of the patients participating in the study were examined by a neurologist, and the exmination included collecting anamnesis (date of surgery, presence of chemotherapy, radiation therapy) and complaints (for edema of the upper limb on the side of surgical treatment, sensitivity disorders of the upper limb, paresthesia, muscle weakness, restriction of movement in the shoulder joint, pain in the upper limb and upper arm, headaches, dizziness, sleep disorders). During the examination, the mobility in the shoulder joint was assessed, the Adson test was used to assess the thoracic outlet syndrome (which consists of pulsing palpitations on the right and left radial artery when turning the head to the right and left with simultaneous deep breathing), and hand dynamometry to assess the strength of the hands on both sides. A comparative measurement of the circumference of the hands was performed at five points to assess the edema. All patients were tested using scales and questionnaires to assess the level of their pain syndrome (VAS scale, McGill questionnaire), the presence of anxiety and depressive disorders (Zung depression scale, STAI anxiety scale), and quality of life assessment (SF-36 questionnaire).

The study was approved by the ethics committee of Almazov National Medical Research Centre (Protocol number 05112019 from 11 November 2019) and was performed in accordance with the Declaration of Helsinki. All subjects provided written informed consent after receiving a complete explanation of the.

### 2.3. Rs-fMRI

Imaging was obtained on a 3 T scanner (Siemens, Germany). Patients underwent an MRI of the brain, which included the standard MRI protocol (using T1-, T2-, TIRM, MPRAGE, DWI) and rs-fMRI (BOLD). The standard MRI protocol was used to exclude the presence of organic brain pathology in patients following BC treatment and in the control group. Patients were asked to remain awake, keep their eyes closed, and lie still.

### 2.4. Data Processing and Statistical Analyses

The postprocessing of the rs-fMRI data was carried out using the CONN v1.7 software package. Statistical processing and evaluation of the neuroimaging study results of the patients with post-treatment symptoms and the control group were carried out using seed-based analysis and independent component analysis (ICA). The Statistica 10 program was also used to analyze the dimensional data. A comprehensive statistical analysis of the study data was carried out. For the statistical description of the measured data, their agreement with the normal distribution and the estimation of the mean values and medians with 95% confidence intervals was verified.

## 3. Results

All of the patients had certain complications after breast cancer treatment, such as lymphedema of the upper limb on the side of surgical treatment (*n* = 23, 50%), sensitivity disorders of the upper limb (*n* = 23, 50%), paresthesia (*n* = 21, 46%), muscle weakness (*n* = 26, 56%), restriction of movement in the shoulder joint (*n* = 19, 41%), pain in the upper limb (*n* = 24, 52%), headaches (*n* = 25, 54%), dizziness (*n* = 18, 39%), and sleep disorders (*n* = 16, 34%). When performing functional tests, a positive Adson test was detected in 24 patients (52%). In 26 out of 46 patients (56%), there was a decrease in hand strength on the side of surgical treatment when performing hand dynamometry.

According to the anxiety scale, 20 patients (44%) showed high situational anxiety, and 27 (60%) showed high personal anxiety. A total of 19 out of 46 patients (41%) were diagnosed with mild depression using the Zung scale. According to the results of the SF-36 quality of life questionnaire, there was a decrease in the overall physical well-being index in 40 patients (88%) and in the overall mental well-being index in 36 patients (80%).

The patients were divided into subgroups depending on the presence of certain clinical syndromes after breast cancer treatment (Table 1).

During the postprocessing of the rs-fMRI data, changes in the functional connections of the medial prefrontal cortex (MPFC) with other parts of the brain were analyzed. The choice of the MPFC as a region of interest in the study is due to its importance as one of the central connectivity hubs of the DMN. MPFC connects extensive areas, including the orbitofrontal cortex and structures, such as the periaqueductal gray matter, the amygdala, and the hypothalamus, and plays an important role in the transmission of somatosensory information to the structures that are responsible for motor and visceral reactions, that participate in the intrinsic reward system, and that are responsible for decision-making [26].

In the current study, an intergroup statistical analysis of the functional connectivity of the DMN between several groups was carried out, where the following comparisons were achieved:(1)Comparison of the connectivity differences between all patients after breast cancer treatment who participated in the study and the control group;(2)Comparison between patients after breast cancer treatment with and without lymphedema;(3)Comparison between patients after breast cancer treatment with the presence of pain in the upper limb and without;(4)Comparison between patients after breast cancer treatment with vestibulocerebellar ataxia and without;(5)Comparison between patients after breast cancer treatment with depression and without depression.

This analysis was carried out in order to assess how a particular syndrome affects the functional connectivity of the DMN and how significant these changes are.

### 3.1. Resting State Functional MRI Results

#### 3.1.1. All Patients after Breast Cancer Treatment in Comparison with Control Group

According to the results of the comparative analysis between the patients after breast cancer treatment (*n* = 46) and the control group (*n* = 20), a decrease in functional connectivity was revealed between the MPFC and the right fusiform gyrus (*p* = 0.046) and in the cortex of the left precentral gyrus (*p* = 0.032) in patients after breast cancer treatment compared to the healthy female volunteers. There was an increase in the functional connectivity between the MPFC and the operculum cortex of the parietal lobes on both sides (*p* = 0.018, *p* = 0.036) (Figure 1; Table 2).

#### 3.1.2. Lymphedema

When comparing functional connectivity in patients after breast cancer treatment with lymphedema and in patients without it, there was a decrease in the connectivity between the MPFC and the occipital cortex on both sides (*p* = 0.012, *p* = 0.013), the cortex of the left middle temporal gyrus (*p* = 0.014), the left fusiform gyrus (*p* = 0.045), and the cortex of the left inferior (*p* = 0.020) and the middle frontal (*p* = 0.022) gyrus. There was an increase in the number of positive connections between the MPFC and the thalami on both sides (*p* = 0.017, *p* = 0.029) and the right cerebellar hemisphere (*p* = 0.036) (Figure 2; Table 3).

#### 3.1.3. Postmastectomy Pain Syndrome

A comparative statistical analysis of the functional connectivity of the patients with postmastectomy pain syndrome compared to patients without postmastectomy pain syndrome showed an increase in the number of negative connections of MPFC with the cortex of the right inferior frontal gyrus (*p* = 0.003), right inferior temporal gyrus (*p* = 0.007), and right amygdala (*p* = 0.046). There was also an increase in the number of negative functional connections between the MPFC and the salience network (*p* = 0.009) as well as the dorsal attention network (*p* = 0.036). Positive connections were found between the MPFC and the pole of the left occipital lobe (*p* = 0.029) and the left cerebellum hemisphere (0.0034) (Figure 3; Table 4).

#### 3.1.4. Vestibulocerebellar Ataxia

A comparative statistical analysis of the functional connectivity in patients after breast cancer treatment with vestibulocerebellar ataxia compared to in patients without it revealed a decrease in the connectivity between the MPFC and the temporo-occipital fusiform cortex on both sides (*p* = 0.023, *p* = 0.032), the lateral occipital cortex on both sides (*p* = 0.016, *p* = 0.049), the left cerebellum (*p* = 0.037), as well as in the right Heschl’s gyrus (*p* = 0.024). There was a significant increase in the number of positive connections of MPFC with the right caudate nucleus (*p* = 0.003) (Figure 4; Table 5).

#### 3.1.5. Depression

In a comparative statistical analysis of the functional connectivity in patients with the presence of depression after breast cancer treatment compared to in patients without depression, there was a decrease in the MPFC connectivity with the left cuneal cortex (*p* = 0.007), the right fusiform gyrus (*p* = 0.028), and the planum polare of the right temporal lobe (*p* = 0.023). There was a bilateral change in the connectivity between the MPFC and the parahippocampal gyrus: there was an increase in the number of positive connections on the right and a decrease in the number of positive connections on the left (*p* = 0.047, *p* = 0.011). There was an increase in the number negative connections in the dorsal attention network (*p* = 0.002) (Figure 5; Table 6).

## 4. Discussion

When analyzing the rs-fMRI data in patients after breast cancer treatment compared to the control group, there was a change in the functional connectivity between the MPFC and a number of significant areas of the brain. In particular, a decrease in the connectivity of the MPFC with the fusiform gyrus was revealed, which was observed both in the main group of patients compared to the control group and in the groups of patients with manifestations of lymphedema, vestibulocerebellar ataxia, and depression. The fusiform gyrus is an important area at the junction of the ventral cortex of the temporal and occipital lobes and is involved in the processes of visual perception, including in the perception of faces, and plays a role in the cognitive processes of memory, attention, and emotions. Changes in the functional connectivity of the fusiform gyrus have been described in patients with amnesic mild cognitive impairment (aMCI) [27] as well as in the pathogenesis of depressive disorders [28,29]. Changes in the functional connectivity between the MPFC and fusiform gyrus in patients with neurological complications after breast cancer treatment may indicate the presence of initial cognitive impairment caused by the surgery, radiation, and/or by chemotherapy treatment. For clarification, the correlation of the obtained neuroimaging data with the results of neuropsychological testing should be assessed.

Currently, it is believed that DMN is active when individuals are engaged in stimulus-independent thought, such as autobiographical memory, and the activation of the DMN decreases during external or attention-demanding tasks that involve mental control [30,31,32]. Therefore, the activation of the DMN is inversely correlated with activation in regions such as the precentral gyrus, which is a part of the somatomotor cortex and is involved in the brain networks that are activated for external tasks that demand attention and mental control [33]. Our study showed the significant decline in the functional connectivity between the MPFC and left precentral gyrus in patients after breast cancer treatment compared to the participants in the control group. These changes in the functional connectivity between the DMN and somatomotor areas could be due to chronic pain syndrome, which can alter DMN–somatomotor cortex connectivity [34].

In our study, there was a change in the functional connectivity between the MPFC and the parahippocampal gyri in the patients with diagnosed mild depressive disorder. The parahippocampal gyrus is one of the most important centers of the DMN and acts as a connectivity hub between such structures as the MPFC and the posterior cingulate cortex on the one hand and the hippocampus on the other [35]. According to recent data, a decrease in the indirect effect of the MPFC on the hippocampus through functional connections with the parahippocampal gyrus may be one of the pathogenetic mechanisms for the development of cognitive deficits [36,37], and an increase in the connectivity between the posterior cingulate cortex and the parahippocampal gyrus plays a role in the occurrence of depressive symptoms [38,39]. Our study also showed a decline in the functional connectivity between the MPFC and the cuneal cortex in patients with depression after breast cancer treatment, which corresponds to previous studies on depressive disorders [40].

In patients with diagnosed postmastectomy pain syndrome in the upper limb and in the postoperative area, a functional reorganization of the DMN was observed and included the involvement of such structures as the cortex of the inferior frontal and middle temporal gyrus and the amygdala. A number of recent studies have shown that the amygdala plays a significant role in the spontaneous activation of “pain neural networks” [41,42] as well as in the pathogenesis of chronic pain syndrome in general, including its emotional and cognitive component [43]. The rs-fMRI data obtained in our study in patients with postmastectomy pain syndrome not only indicate the excessive activation of the structures of the “pain connectome”, but also a violation of the functional regulation between the structures of the DMN and the salience network, the main centers of which are the anterior insular and anterior cingulate cortex. The violation of connectivity between these two neural networks is considered one of the key aspects of the centralization of chronic pain syndrome [44]. It is known that the transcranial magnetic stimulation (TMS) method is used for chronic pain syndromes of various etiologies [45]. Knowledge about the structure of the functional disorganization of the “pain connectome” makes it possible to influence specific areas of the brain using TMS in patients to achieve a better effect of reducing the frequency and the intensity of the pain. Taking into account the complexity of the organization of the functional networks of the brain involved in the pathogenesis of pain syndrome, it may be necessary to develop new approaches for the application of the TMS method.

A number of patients participating in the study had symptoms of cerebellar ataxia in the form of dizziness, instability in the Romberg pose, and during gait changes. When comparing the rs-fMRI data in this group of patients with data from the group of patients who did not have vestibulocerebellar ataxia, there was an alteration in the connectivity of the MPFC with the left hemisphere of the cerebellum, which indicates the presence of cerebellar disorders in the pathogenesis of post-mastectomy syndrome in these patients. These changes in DMN–cerebellar connectivity may be caused by a chronic vertebral-basilar insufficiency in patients as a long-term effect of breast cancer surgery and/or radiation therapy, such as local fibrous-atrophic changes in the postoperative area and hypertrophy of the scalenus muscles, which leads to the spasming of the vertebral artery on the treatment side [46,47]. This assumption requires further study and comparison with data from other imaging techniques, such as ultrasounds of the arteries in the neck [48].

The most vivid neuroimaging picture obtained during rs-fMRI was observed in patients with the presence of upper limb lymphedema after breast cancer treatment. In this group of patients, there were significant changes in the connectivity of the DMN in comparison to the patients without the presence of lymphedema, with the overall decline in the functional connectivity of the DMN and a more pronounced decline in the connectivity between the MPFC and inferior and middle frontal gyri cortex, middle temporal gyrus, and lateral occipital cortex on the left. These results may indicate a significant role of lymphedema in the pathogenesis of neurological disorders in patients after surgical treatment for breast cancer and may be due to a complex of vascular and neurodegenerative changes [49]. However, it should be noted that the development of lymphedema occurs more often in women with a more severe stage of the disease, and the effects of chemo- and radiation therapy should not be excluded from the pathogenesis of neuronal damage [50,51]. Nevertheless, fMRI can be used as a helpful tool for diagnosing neurological disorders in women after breast cancer treatment and even for the prediction of the long-term cognitive deficits [52].

## 5. Conclusions

In conclusion, the application of rs-fMRI provides a possibility to assess functional changes in the brains of patients following breast cancer treatment that occur against the background of lymphedema, postmastectomy pain syndrome, vertebrobasilar insufficiency, anxiety, and depressive disorders. The variety of clinical symptoms, diagnostic difficulties, multi-system complications of breast cancer treatment, and significant decreases in the quality of life determine the need for a comprehensive diagnostic (including method for assessing functional changes in the brain—rs-fMRI), treatment, and rehabilitation approaches for this group of patients.

## Figures and Tables

**Figure 1 jcm-11-00617-f001:**
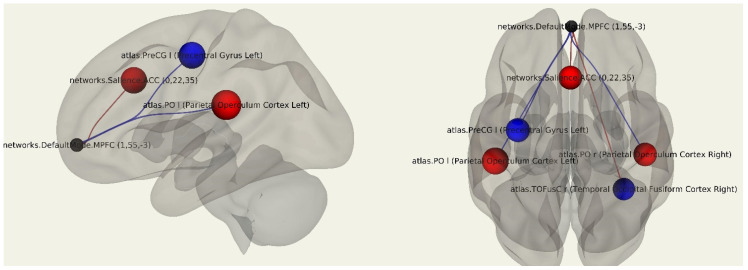
Three-dimensional reconstruction of the functional connections between the MPFC and various areas of the brain in a group of patients after breast cancer treatment using seed-based analysis. Positive functional connections between the MPFC and the zones of interest are indicated in red, and negative ones are indicated in blue.

**Figure 2 jcm-11-00617-f002:**
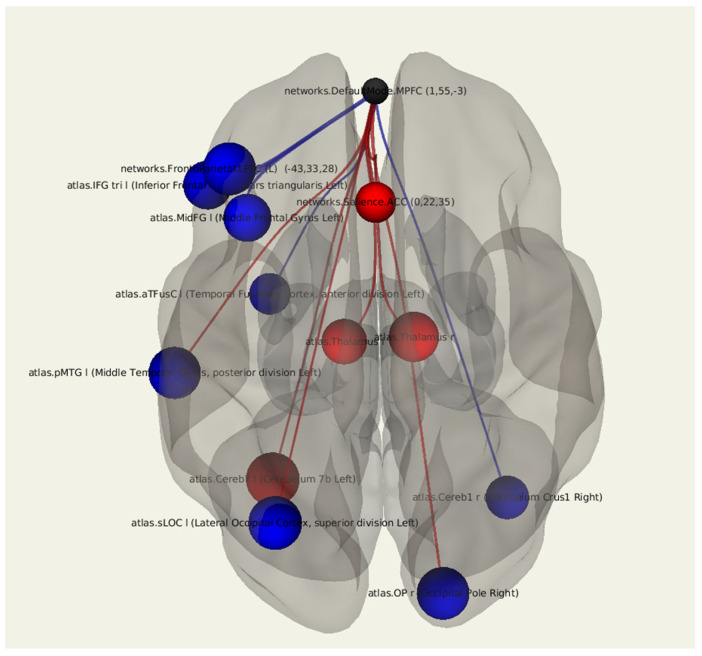
Three-dimensional reconstruction of the functional connections between the MPFC and various areas of the brain in a group of patients after breast cancer treatment with the presence of lymphedema using seed-based analysis. Positive functional connections between the MPFC and the zones of interest are indicated in red, and negative ones are indicated in blue.

**Figure 3 jcm-11-00617-f003:**
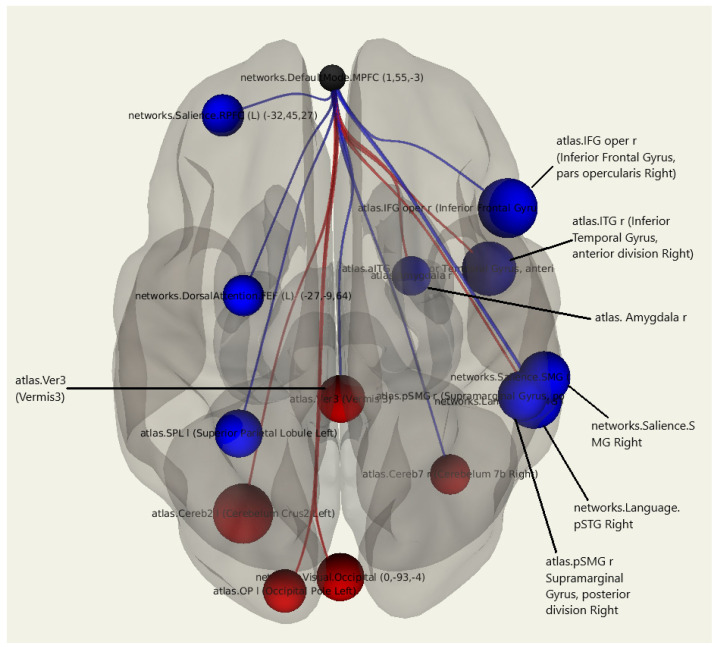
Three-dimensional reconstruction of the functional connections between the MPFC and various areas of the brain in a group of patients with postmastectomy pain syndrome using seed-based analysis. Positive functional connections between the MPFC and the zones of interest are indicated in red, and negative ones are indicated in blue.

**Figure 4 jcm-11-00617-f004:**
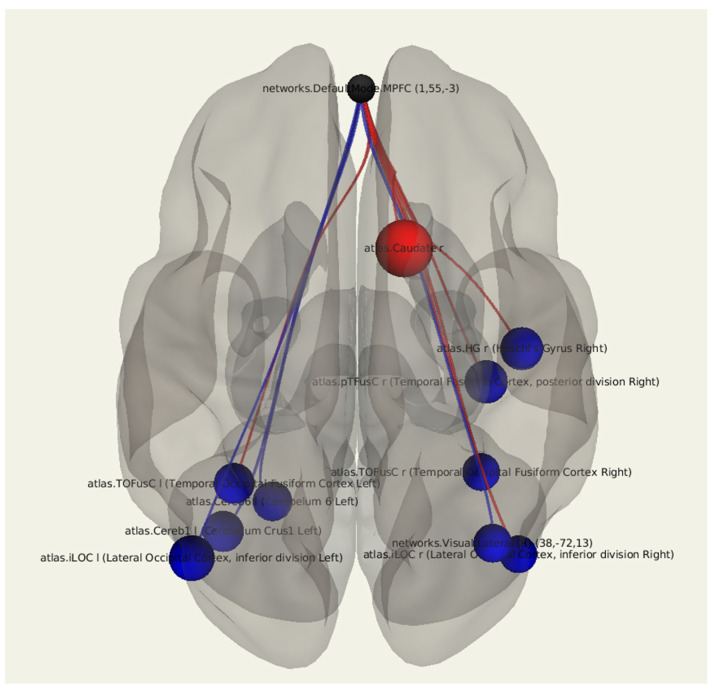
Three-dimensional reconstruction of the set of functional connections between the MPFC and various brain areas of patients after breast cancer treatment with the presence of vestibulocerebellar ataxia using seed-based analysis. The positive functional connections between the MPFC and the zones of interest are indicated in red, and the negative ones are indicated in blue.

**Figure 5 jcm-11-00617-f005:**
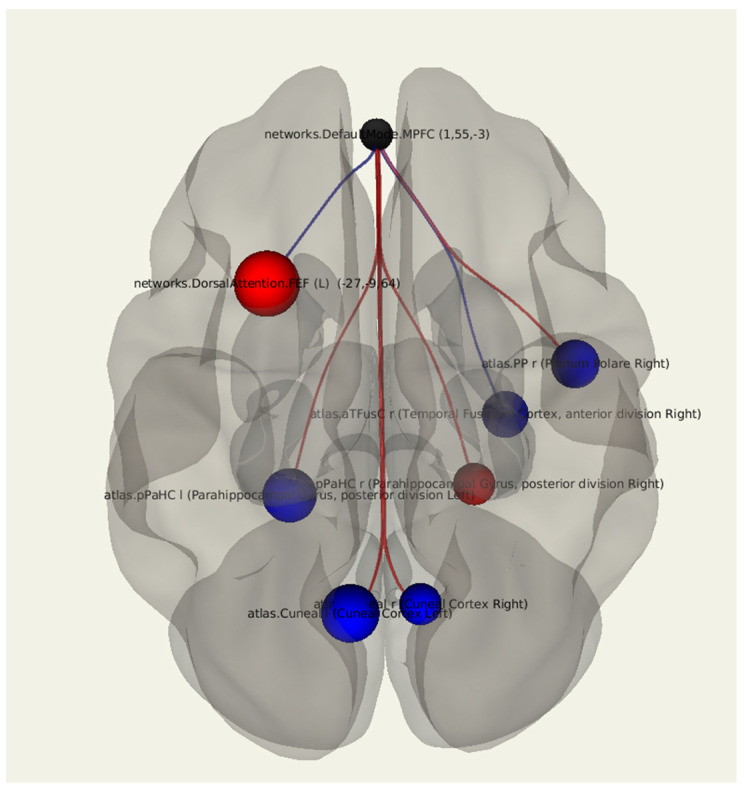
Three-dimensional reconstruction of the set of functional connections between the MPFC and various brain areas of patients with the presence of depression after breast cancer treatment using seed-based analysis. Positive functional connections between the MPFC and the zones of interest are indicated in red, and negative ones are indicated in blue.

**Table 1 jcm-11-00617-t001:** The number of patients in groups depending on syndrome.

Syndrome	Number of Patients with the Syndrome	Number of Patients without the Syndrome
Lymphedema	23	23
Postmastectomy pain syndrome	24	22
Vestibulocerebellar ataxia	18	28
Depression	19	27

**Table 2 jcm-11-00617-t002:** Main regions of interest with MPFC connections in the group of patients after breast cancer treatment.

Target Region	Side	T	Beta	*p*-unc
Parietal Operculum	Left	2.43	0.11	0.018390
Precentral Gyrus	Left	−2.20	−0.11	0.032464
Parietal Operculum	Right	2.15	2.15	0.036042
Fusiform Gyrus (Temp-Occ)	Right	−2.04	−2.04	0.046336

**Table 3 jcm-11-00617-t003:** Main regions of interest where MPFC connections were present in patients after breast cancer treatment with lymphedema.

Target Region	Side	T	Beta	*p*-unc
Lateral Occipital Cortex	Left	−2.74	−0.23	0.012076
Cerebellum	Left	2.73	0.21	0.012222
Occipital Pole	Right	−2.69	−0.18	0.013510
Middle Temporal Gyrus	Left	−2.66	−0.28	0.014181
Thalamus	Right	2.57	0.20	0.017496
Inferior Frontal Gyrus	Left	−2.49	−0.24	0.020691
Middle Frontal Gyrus	Left	−2.46	−0.23	0.022441
Thalamus	Left	2.33	0.16	0.029463
Cerebellum	Right	−2.22	−0.18	0.036737
Fusiform Gyrus (Temp)	Left	−2.12	−0.15	0.045543

**Table 4 jcm-11-00617-t004:** Main regions of interest where MPFC connections were present in patients with postmastectomy pain syndrome.

Target Region	Side	T	Beta	*p*-unc
Cerebellum	Left	3.34	0.24	0.003469
Inferior Frontal Gyrus	Right	−3.32	−0.26	0.003615
Inferior Temporal Gyrus	Right	−3.02	−0.21	0.007069
Salience network (SMG)	Right	−2.88	−0.29	0.009688
Occipital Pole	Left	2.36	0.18	0.029258
Dorsal Attention. FEF		−2.25	−0.17	0.036340
Amygdala	Right	−2.13	−0.14	0.046265

**Table 5 jcm-11-00617-t005:** Main regions of interest where MPFC connections were present in patients after breast cancer treatment with vestibulocerebellar ataxia.

Target Region	Side	T	Beta	*p*-unc
Caudate	Right	3.14	0.28	0.003531
Lateral Occipital Cortex	Left	−2.51	−0.25	0.016856
Fusiform Gyrus (Temp)	Right	−2.38	−0.19	0.023308
Heschl’s Gyrus	Right	−2.36	−0.19	0.024008
Fusiform Gyrus (Temp-Occ)	Left	−2.23	−0.17	0.032363
Cerebellum	Left	−2.16	−0.16	0.037966
Lateral Occipital Cortex	Right	−2.04	−0.19	0.049005

**Table 6 jcm-11-00617-t006:** Main regions of interest where MPFC connections are present in patients with depression after breast cancer treatment.

Target Region	Side	T	Beta	*p*-unc
Dorsal Attention. FEF		3.39	−0.18	0.002925
Cuneal Cortex	Left	−2.99	−0.21	0.007221
Parahippocampal gyrus	Left	−2.77	−0.16	0.011720
Planum Polare	Right	−2.46	−0.16	0.023100
Fusiform Gyrus (Temp)	Right	−2.37	0.13	0.028196
Parahippocampal Gyrus	Right	2.11	−0.14	0.047445

## Data Availability

Not applicable.

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
