# Peer review of "Functional Network Connectivity Reveals the Brain Functional Alterations in Breast Cancer Survivors"

_jcm, 2022, doi:10.3390/jcm11030617_

Round 1

Reviewer 1 Report

Dear authors

Thank you for your manuscript submission to our journal. I enjoyed your research work but felt many problems in your manuscript as follows;

Major points

1.You assessed the breast cancer patients with post-mastectomy pain syndrome using rs-fMRI. But your only enrolled 24 patients with pain in the upper limb/postoperative area.  You should analyze the data obtained from these 24 patients. 

2.You should at least suggest a clue to alleviate the pain through this study.

Minor points

Concerning the expressions about breast cancer, you should revise many points.

  1. Local (surgery and /or radiation therapy)→local( surgery ± radiation therapy)
  2. Removing the small and / or large muscles→Removing the small muscle
  3. Modified Patey radical mastectomy→Patey radical mastectomy

Author Response

We would like to thank the reviewer for the provided comments. We have revised the manuscript accordingly. Below we provide the answers to the raised questions.

Comment (1): You assessed the breast cancer patients with post-mastectomy pain syndrome using rs-fMRI. But your only enrolled 24 patients with pain in the upper limb/postoperative area.  You should analyze the data obtained from these 24 patients. 

Answer (1): The analysis of these group of patients is incorporated into the section 3.1.3.

Comment (2): You should at least suggest a clue to alleviate the pain through this study.

Answer (2): In the Discussion section we have included the paragraph describing the possible application of transcranial magnetic stimulation (TMS) that could be employed for elevation of pain. Additionally, we included a new reference [45].

Comment (3): 

Concerning the expressions about breast cancer, you should revise many points.

  1. Local (surgery and /or radiation therapy)→local( surgery ± radiation therapy)
  2. Removing the small and / or large muscles→Removing the small muscle
  3. Modified Patey radical mastectomy→Patey radical mastectomy

Answer (3): These were corrected in the manuscript.

Reviewer 2 Report

The Authors describe a series of 46 patients with post-mastectomy syndrome in which the rs-fMRI was used to assess functional changes in the brain. The work is well conducted and allows the identification of changes in functional connectivity in the brain. These results could be useful to assess and monitor patients that need rehabilitation.  

 the need for a comprehensive diagnostic (including method for assessing 348 functional changes in the brain - rs-fMRI), treatment and rehabilitation approach to this 349 group of patients 

Author Response

We would like to thank the reviewer for the provided comments. Herein, we submit a revised version of the manuscript where we have incorporated the changes.

Comment (1): 

The Authors describe a series of 46 patients with post-mastectomy syndrome in which the rs-fMRI was used to assess functional changes in the brain. The work is well conducted and allows the identification of changes in functional connectivity in the brain. These results could be useful to assess and monitor patients that need rehabilitation.  

 the need for a comprehensive diagnostic (including method for assessing 348 functional changes in the brain - rs-fMRI), treatment and rehabilitation approach to this 349 group of patients 

Answer (1): We would like to thank the reviewer for this notion. Currently, on the basis of the rs-fMRI we are developing a clinical trial to monitor these group of patients and elaborate a treatment protocol.

Round 2

Reviewer 1 Report

Dear authors

Thank you for your manuscript re-submission to our journal. I’ve read your revised manuscript and felt marked improvement except for one point.

Line 29 and 106

In your original manuscript, you described “the late postoperative period (>6 months)”. But in your revised manuscript, you changed the term from >6 months to >12 months. Why did the change occur? 

Author Response

We would like to thank the reviewer for the provided comment. Below we provide the answer to the raised question.

Comment (1): In your original manuscript, you described “the late postoperative period (>6 months)”. But in your revised manuscript, you changed the term from >6 months to >12 months. Why did the change occur? 

Answer (1): Initially, it was planned to include patients in the study in the late postoperative period (more than 6 months), but when re-analyzing the database, it was found that all the patients included in the study were in the late postoperative period (more than 12 months after mastectomy), which served as the basis for changing the term in the inclusion criteria.